# Brain Proteome-Wide Association Study Identifies Candidate Genes that Regulate Protein Abundance Associated with Post-Traumatic Stress Disorder

**DOI:** 10.3390/genes13081341

**Published:** 2022-07-27

**Authors:** Zhen Zhang, Peilin Meng, Huijie Zhang, Yumeng Jia, Yan Wen, Jingxi Zhang, Yujing Chen, Chun’e Li, Chuyu Pan, Shiqiang Cheng, Xuena Yang, Yao Yao, Li Liu, Feng Zhang

**Affiliations:** Key Laboratory of Trace Elements and Endemic Diseases of National Health and Family Planning Commission, School of Public Health, Health Science Center, Xi’an Jiaotong University, Xi’an 710000, China; zhang785334865@yeah.net (Z.Z.); mengpeilin@stu.xjtu.edu.cn (P.M.); zhj2020@stu.xjtu.edu.cn (H.Z.); jiayumemg@mail.xjtu.edu.cn (Y.J.); wenyan@mail.xjtu.edu.cn (Y.W.); 3120315071@stu.xjtu.edu.cn (J.Z.); c18003409402@163.com (Y.C.); lichune@stu.xjtu.edu.cn (C.L.); panchuyu_dsa@163.com (C.P.); chengsq0701@stu.xjtu.edu.cn (S.C.); smile940323@stu.xjtu.edu.cn (X.Y.); yao3077690800@stu.xjtu.edu.cn (Y.Y.); liuli0624@stu.xjtu.edu.cn (L.L.)

**Keywords:** post-traumatic stress disorder, human brain proteome, proteome-wide association studies, transcriptome-wide association studies, genetics

## Abstract

Although previous genome-wide association studies (GWASs) on post-traumatic stress disorder (PTSD) have identified multiple risk loci, how these loci confer risk of PTSD remains unclear. Through the FUSION pipeline, we integrated two human brain proteome reference datasets (ROS/MAP and Banner) with the PTSD GWAS dataset, respectively, to conduct a proteome-wide association study (PWAS) analysis. Then two transcriptome reference weights (Rnaseq and Splicing) were applied to a transcriptome-wide association study (TWAS) analysis. Finally, the PWAS and TWAS results were investigated through brain imaging analysis. In the PWAS analysis, 8 and 13 candidate genes were identified in the ROS/MAP and Banner reference weight groups, respectively. Examples included *ADK* (*p*_PWAS-ROS/MAP_ = 3.00 × 10^−5^) and *C3orf18* (*p*_PWAS-Banner_ = 7.07 × 10^−31^). Moreover, the TWAS also detected multiple candidate genes associated with PTSD in two different reference weight groups, including *RIMS2* (*p*_TWAS-Splicing_ = 3.84 × 10^−2^), *CHMP1A* (*p*_TWAS-Rnaseq_ = 5.09 × 10^−4^), and *SIRT5* (*p*_TWAS-Splicing_ = 4.81 × 10^−3^). Further comparison of the PWAS and TWAS results in different populations detected the overlapping genes: *MADD* (*p*_PWAS-Banner_ = 4.90 × 10^−2^, *p*_TWAS-Splicing_ = 1.23 × 10^−2^) in the total population and *GLO1*(*p*_PWAS-Banner_ = 4.89 × 10^−3^, *p*_TWAS-Rnaseq_ = 1.41 × 10^−3^) in females. Brain imaging analysis revealed several different brain imaging phenotypes associated with *MADD* and *GLO1* genes. Our study identified multiple candidate genes associated with PTSD in the proteome and transcriptome levels, which may provide new clues to the pathogenesis of PTSD.

## 1. Introduction

Post-traumatic stress disorder (PTSD) is a highly debilitating mental disease characterized by persistent cognitive–affective dysfunction [1], which primarily occurs after exposure to a traumatic event. It has been reported that about 10 to 20% of individuals exposed to a traumatic stressor will suffer from PTSD [2], and the lifetime prevalence is about 7% in the general population [3]. Moreover, other studies have found gender differences in PTSD; compared with males, females have a higher PTSD incidence rate, more severe symptoms, and more chronic symptoms [4]. PTSD not only causes lower academic achievement and increased rates of suicide attempts and substance abuse [5], but also leads to a variety of comorbid mental disorders, such as major depression and generalized anxiety disorder [6]. Because childhood trauma and post-traumatic stress disorder place a burden on society and families in terms of health care utilization and economic expenditure [5], it is necessary to explore the etiology and pathogenesis of PTSD.

Previous studies have mainly focused on the role of the prefrontal cortex (PFC)–hippocampal–amygdala network in mediating PTSD [7,8] and determined that dysfunction in these brain regions underlies the development of PTSD. The PFC, hippocampus, and amygdala are the centers of the brain networks that mainly mediate fear learning and memory processes, and when the neural circuits supporting them are dysfunctional, fear learning and memory processes will be disturbed in PTSD [9]. Imaging studies have supported this hypothesis by revealing abnormalities in both the hippocampus and medial PFC in the brains of PTSD patients [10]. In addition, gender differences have been found in PTSD studies, including gender roles, genetic predisposition, and hormonal influences [11]. Gender-specific PTSD vulnerability was partially mediated by gender differences in the fear system [4], which further illustrated the possibility that PTSD is associated with specific brain regions. Although a series of molecular biology, neuroimaging, and genetic studies have provided useful information about the role of the PFC–hippocampal–amygdala network in the development of PTSD, its pathogenesis remains unclear.

It is worth mentioning that genetic factors play an important role in the occurrence and development of PTSD. For example, twin studies have suggested genetic risk factors for PTSD development, with the heritability of PTSD symptoms generally estimated at about 30% [12]. Numerous genome-wide association studies (GWASs) of PTSD have revealed multiple genetic risk loci that are significantly associated with PTSD [13,14]. For example, the *NLGN1* gene is localized in excitatory synapses and plays an important role in learning and memory; knockout of *NLGN1* can lead to loss of fear memory storage, and variation in *NLGN1* may predispose individuals to experiencing higher levels of anxiety and fear, potentially increasing their risk of developing PTSD after a traumatic event [13]. In addition, a transcriptome-wide association study (TWAS) has suggested the association between PTSD and abnormal gene expression in the brain and peripheral blood cells [15]. These studies have explored the pathogenesis of PTSD from the perspectives of genetics, epigenetics, and transcription factors, but rarely has research directly explored the correlation between the brain proteome and PTSD.

With the continuous improvement of GWAS methods and the emergence of high-throughput proteome sequencing technology in complex tissues, the GWAS promoted the application of the proteome-wide association study (PWAS) and TWAS in complex human diseases. A PWAS is a protein-centric approach to genetic association research that considers the proteomic context of genetic variation and its functional effects and decomposes variant-level associations into individual gene values (determined by the association between variants and protein levels), then evaluates the combined association of these variants with outcomes [16]. For example, Wingo et al. identified 19 genes in the PWAS analysis that were consistent with causality in depression and functioned through *cis*-regulated abundance of their respective brain proteins [17]. The TWAS is commonly used to identify genes whose expression is significantly associated with complex traits without directly measuring their expression levels. Specifically, the TWAS uses genotype and expression data from an external reference panel to determine the association between the genetic variant and gene expression, then identifies genes whose *cis*-regulated expression is associated with diseases or phenotypes by integrating gene expression data with genome-wide associations from a large-scale GWAS [18]. At present, the TWAS is widely applied in the study of psychiatric disorders. For example, a TWAS identified 26 risk genes whose *cis*-regulated expression was significantly associated with anxiety [19]. Although a large proportion of variation in complex human traits is caused by genetic factors, the mechanisms between genetic factors and traits are generally difficult to understand. The reason is that complex traits or diseases can result from pathological changes at any step in the gene expression process (from transcription to translation to protein function, or reverse transcription). Previous studies have found that many genetic factors can affect complex traits or diseases by modulating gene expression, finally altering protein abundance levels [20]. Therefore, we hypothesized that specific genetic variations affect the pathogenesis of PTSD by altering gene expression levels of proteins and transcriptomes.

In our study, based on the PTSD GWAS database, we analyzed the total population and different gender populations through PWAS and TWAS analysis, respectively. Specifically, the FUSION pipeline was used to integrate PTSD gene expression data with different brain proteome and transcriptome data to identify the genes associated with PTSD at the protein and mRNA levels, respectively. Then, brain imaging analysis was performed to identify brain regions that may be associated with PTSD. By combining GWAS, TWAS and PWAS datasets, we hope to identify genetic loci associated with PTSD in gene expression, providing new insights for the pathogenesis of PTSD.

## 2. Materials and Methods

### 2.1. GWAS Dataset for Post-Traumatic Stress Disorder

The GWAS dataset for PTSD used in our study was based on summary statistics from the Psychiatric Genome Consortium-Post-Traumatic Stress Disorder Group (PGC-PTSD) [21]. It is a multi-ethnic cohort of more than 30,000 PTSD cases and 170,000 controls. The diagnosis of PTSD relied on various tools and different versions of the DSM. In addition, quality control was carried out using the PGC’s RICOPILI pipeline in accordance with sample exclusion criteria: using single nucleotide polymorphisms (SNPs) with call rates >95% and samples with call rates <98%, excluding deviation from the expected inbreeding coefficient (f_het_ <−0.2 or >0.2) or a sex discrepancy between reported and estimated sex based on inbreeding coefficients calculated from SNPs on X chromosomes. Genotyping was performed using a series of Illumina genotyping arrays and Affymetrix Axiom arrays, and imputation used IMPUTE2 based on 1000 Genomes phase 3 data (1KGP phase 3). The within-study relatedness in GWAS research was estimated by the IBS function in PLINK 1.9.

It is worth noting that the PGC-PTSD not only carried out a GWAS analysis of PTSD in the total population, but also compiled GWAS summary statistics on PTSD in the by-sex population. We selected only data from European ancestry including 1,161,480 significantly independent SNPs in the total population, and 1,155,985 and 1,169,775 significantly independent SNPs in the male and female populations, respectively. Detailed description of the sample information, experimental design, and analysis methods are available in published studies [21].

### 2.2. Human Brain Proteome Reference Weight for PWAS

In our PWAS analysis of PTSD, two human brain proteome reference weight data were obtained from published studies, including the Religious Orders Study and Rush Memory and Aging Project (ROS/MAP) [22] and the Banner Sun Health Research Institute (Banner) [23]. According to a previous study of the human brain proteome [24], the proteome was sequenced and combined with significant SNPs in the GWAS database to identify *cis*-regulated proteins that were significantly related to genetic variation. Finally, these proteins were used as reference weights for subsequent PWAS analysis. Quality control criteria carried out for protein reference weight data were as follows: we excluded proteins with a missing value (global internal standards were used to check for proteins outside the 95% confidence interval and set to missing); each protein abundance was scaled to sample a specific total protein abundance to eliminate the effect of protein loading differences; and outlier samples were identified and removed by iterative principal component analysis (in each iteration, samples with more than four standard deviations were removed from the mean of the first or second principal component). Finally, protein abundance was normalized by the residual of the linear regression to eliminate the effects of protein batch, sex, age at death, and post-mortem interval. Detailed information on proteome sequencing, quality control, and standardization can be obtained from the study by Wingo et al. [24]. After analysis and quality control, the ROS/MAP proteome (*N* = 376) and the Banner proteome (*N* = 152) identified 1475 and 1139 proteins, respectively, that showed a significant *cis*-association with genetic variation.

### 2.3. Proteome-Wide Association Study (PWAS)

Using the FUSION pipeline (http://gusevlab.org/projects/fusion/ accessed on 20 December 2021), we integrated the PTSD GWAS dataset with two different reference human brain proteomes (ROS/MAP [25] and Banner [21]) to perform the PWAS analysis [18]. Specifically, in the FUSION pipeline, the protein weights of the two human brain proteomes, ROS/MAP and Banner, were respectively estimated (the two analysis processes were consistent); then we calculated the PTSD genetic effect (PWAS z-score) and combined it with the pre-calculated brain proteome reference weight (z-score × proteome weight) to evaluate the effects of significant SNPs in the GWAS on the protein abundance. Finally, FUSION identified candidate genes associated with PTSD regulating the abundance of proteins in the brain. The default settings and parameters recommended by FUSION were used in the analysis. (The main operating parameters were as follows: --sumstats; --weights; --weights_dir; --ref_ld_chr; --chr; --out; --perm; --GWASN. Specific information on these parameters can be queried in the FUSION pipeline.) To control the potential effect of multiple testing on the study results, the permutation-based *p*-value (named PERM.ANL) was calculated for each gene by FUSION. The permutation-based *p*-value (*p*_PWAS_) <0.05 was used as the significance threshold in our PWAS analysis.

### 2.4. Transcriptome-Wide Association Study (TWAS)

The TWAS analysis was also conducted based on the FUSION pipeline and was similar to the PWAS analysis in principle and parameters. Unlike the PWAS analysis, the external expression reference panel used for the TWAS analysis was based on mRNA expression level. Briefly, in the FUSION pipeline, we first calculated the reference weights of Rnaseq and Splicing human brain gene expression panels through the multiple prediction models running in FUSION and selected the best reference weights; then we calculated the PTSD genetic effects (TWAS z-score) and combined them with the gene expression weights (z-score × weight) for the TWAS analysis to evaluate the association between gene expression levels and PTSD and to identify candidate genes (based on gene expression levels) associated with PTSD [26]. The default settings and parameters recommended by FUSION were used in the TWAS analysis. In our TWAS analysis, a permutation-based *p*-value (named PERM.ANL, *p*_TWAS_) <0.05 calculated by FUSION was used as the significance threshold.

The gene expression data used in our TWAS analysis were derived from two human brain gene expression reference weights (Rnaseq and Splicing) provided by the FUSION pipeline. Both datasets were obtained from the dorsolateral prefrontal cortex (dPFCS) of 452 European individuals collected by the CommonMind Consortium, and detailed data information is available in the FUSION pipeline.

### 2.5. Brain Imaging Analysis

Utilizing the Oxford Brain Imaging Genetics Server -BIG40 (https://open.win.ox.ac.uk/ukbiobank/big40/ accessed on 22 December 2021), we explored the brain imaging traits associated with the overlapping genes (*MADD* and *GLO1*) identified by the PWAS and TWAS analyses. Based on UK Biobank’s multi-model brain imaging data, the site contained the GWAS results of 3935 brain image-derived phenotypes, which can be used to identify brain imaging phenotypes associated with target SNPs and genes and to find specific brain regions by the description of the phenotypes [27]. The analysis used minor allele frequency (MAF) >0.01 as filter. To avoid multiple testing correction, we used multiple testing corrected *p*-values (0.05/3935 = 1.27 × 10^−5^) as a suggestive association threshold; to be conservative, we additionally adopted the more stringent genome-wide significance threshold (*p* = 5 × 10^−8^) as the significant association threshold.

## 3. Results

### 3.1. PWAS Analysis Results

In the PWAS analysis combined with the ROS/MAP proteome reference weights, a total of eight candidate genes associated with PTSD were identified, such as *ADK* (*p*_PWAS-ROS/MAP_ = 3.00 × 10^−5^) in the all population; *PCYOX1* (*p*_PWAS-ROS/MAP_ = 2.70 × 10^−4^) in females; and *BCAN* (*p*_PWAS-ROS/MAP_ = 8.14 × 10^−4^) in males.

Another PWAS analysis combined with Banner proteome reference weights identified 13 candidate genes significantly associated with PTSD, such as *MADD* (*p*_PWAS-Banner_ = 4.9 × 10^−2^) in the all population; *GLO1* (*p*_PWAS-Banner_ = 4.89 × 10^−3^) in females; and *C3orf18* (*p*_PWAS-Banner_ = 7.07 × 10^−31^) in males. Notably, *C3orf18* (*p*_PWAS-all_ = 4.85 × 10^−44^; *p*_PWAS-male_ = 7.07 × 10^−31^) and *ANKRD16* (*p*_PWAS-all_ = 6.85 × 10^−3^; *p*_PWAS-male_ = 3.58 × 10^−3^) overlapped in the all population and male population in the Banner proteome reference weight group. Table 1 summarizes the detailed results of the PWAS analysis.

### 3.2. TWAS Analysis Results

For the TWAS analysis, 35 candidate genes were identified in the Rnaseq gene expression reference weight group, and 53 candidate genes were identified in the Splicing gene expression reference weight group. Among these candidate genes, the Rnaseq reference weight group identified two common genes, *PCDHA9* (*p*_TWAS-All_ = 5.09 × 10^−3^, *p*_TWAS-Female_ = 4.17 × 10^−4^) and *C1orf54* (*p*_TWAS-All_ = 4.28 × 10^−3^, *p*_TWAS-Female_ = 5.15 × 10^−3^). In addition, six common genes were found in the Splicing reference weight group, including *SLC6A7* (*p*_TWAS-All_ = 1.77 × 10^−3^, *p*_TWAS-Female_ = 4.47 × 10^−4^), *GLRX* (*p*_TWAS-All_ = 2.31 × 10^−3^, *p*_TWAS-Female_ = 2.30 × 10^−3^), *AKAP9* (*p*_TWAS-All_ = 1.51 × 10^−13^, *p*_TWAS-Male_ = 1.19 × 10^−10^), and *UNC50* (*p*_TWAS-All_ = 1.51 × 10^−13^, *p*_TWAS-Male_ = 8.88 × 10^−8^). Table 2 summarizes the top five candidate genes significantly associated with PTSD in the TWAS analysis. Detailed results are shown in Appendix A.

### 3.3. Comparison of the PWAS and TWAS Analysis Results

After comparing the results of the PWAS and TWAS analyses in different populations, we found an overlapping gene, *MADD* (*p*_PWAS-Banner_ = 4.90 × 10^−2^, *p*_TWAS-Splicing_ = 1.23 × 10^−2^), in the all population and an overlapping gene, *GLO1* (*p*_PWAS-Banner_ = 4.89 × 10^−3^, *p*_TWAS-Rnaseq_ = 1.41 × 10^−3^), in the female population, but no overlapping gene was detected in the male population. Considering the influence of different population groups, we used the permutation-based *p*-value (0.05/3 = 0.017) corrected by multiple testing as the threshold in the comparison results. *p*-value < 0.017 indicated significant association, and 0.017 < *p*-value < 0.05 indicated suggestive association. Details are shown in Table 3.

### 3.4. Brain Imaging Analysis Result

We further explored the brain imaging traits based on the overlapping genes (*MADD, GLO1)* identified by both the PWAS and TWAS. *MADD* was significantly associated with 19 brain imaging phenotypes, which were described as related to the corresponding brain regions, such as putamen, pallidum, left superior parietal, right cingulate gyrus, hippocampus, and internal capsule brain regions. *GLO1* was suggestively associated with four brain imaging phenotypes, which were described for brain regions such as the right bankssts area, right lateral–orbito-frontal brain regions and functional connectivity. Table 4 lists the results of the ten main brain imaging analyses and descriptions, and complete information is provided in Appendix A.

## 4. Discussion

Recently, PWAS and TWAS analyses have enjoyed a wide application in neurological disorders, providing etiological clues to psychiatric disorders. Examples include a PWAS that identified four common psychiatric disorders and 61 genes (including 48 schizophrenia genes, 12 bipolar disorder genes, 5 depression genes, and 2 attention deficit hyperactivity disorder genes) associated with the risk of psychiatric disorders through regulated protein abundance levels [28]. A TWAS of major depression (MD) identified 94 transcriptome genes that were significantly associated with MD and were differentially expressed in multiple tissues, which indicated that extensive transcriptome changes occur in depression [29]. However, few previous studies have combined the brain proteome with the transcriptome for PTSD. In order to identify candidate genes associated with PTSD within the human brain proteome and transcriptome range, we performed PWAS and TWAS analyses, respectively, on the largest currently available PTSD GWAS dataset. Then, brain imaging analysis was used to further understand the possible brain regions and biological processes of genes associated with PTSD. Through PWAS and TWAS analyses, our study identified multiple significant candidate genes associated with PTSD at the human proteome and transcriptome levels.

Regarding the overlapping genes *MADD* and *GLO1* detected in the PWAS and TWAS analyses, previous studies found that they are not only associated with neurological diseases but also have a certain impact on neurological functions. Specifically, *MADD* (MAP kinase-activating death domain), an enzyme protein-coding gene, encodes a mitogen-activated protein kinase (MAPK)-activating death domain protein, which is mainly expressed in brain tissue and has various cellular functions such as vesicle trafficking, tumor necrosis factor-α (TNF-α)-induced signal transduction, and prevention of cell apoptosis [30]. Schneeberger et al. revealed the role of the *MADD* gene in regulating the physiological functions of the sensory and autonomic nervous system; deletion and mutation of the *MADD* gene can lead to neurodevelopmental, intelligence, and language disorders by affecting TNF-α-dependent signaling pathways and vesicular trafficking [30]. Jun Miyoshi et al. validated the function of *MADD* in the nervous system and found that *MADD* is associated with Alzheimer’s disease (AD) [31]. Another study proved that the downregulation of *MADD* is associated with neuronal cell death in the brain and hippocampal neurons of AD [32]. Moreover, an integrative analysis based on an insomnia GWAS dataset and brain region-related enhancer maps indicated that the *MADD* gene was significantly associated with insomnia, revealing its role in insomnia and other mental disorders [33]. Among the confirmed neurological functions associated with it, *MADD* can cause abnormalities in the hippocampus and neurons, which is thought to underlie the development of PTSD, further enhancing the possibility that *MADD* is related to the pathogenesis of PTSD [34]. As an enzyme protein-coding gene, *GLO1* encodes glyoxalase I and is involved in detoxification of methylglyoxal (MG) in the cellular glycolysis pathway [35]. Previous studies have found that *GLO1* expression and MG accumulation in the brain are associated with the pathogenesis of psychiatric disorders, such as anxiety disorder, depression, autism, and schizophrenia [36]. A recent hospital-based case-control study found significant differences in the distribution of *GLO1* variation, RNA expression, and enzyme activity between schizophrenia patients and controls, suggesting *GLO1* is associated with dysfunction in the left middle frontal gyrus in schizophrenia [37]. However, a link between *GLO1* and PTSD has not been directly demonstrated. Based on existing studies, we speculated that *GLO1* may be involved in the pathogenesis of PTSD by affecting the brain regions associated with PTSD. Further studies are needed to explore the association between *GLO1* and PTSD. The overlap between the PWAS and TWAS results highlighted the consistency in protein and gene expression levels and validated the possibility that overlapping genes are associated with the pathogenesis of PTSD from different perspectives.

Among the significant candidate genes identified by the PWAS analysis of the total population, several genes associated with neurological diseases and nervous system development were also found, such as *ADK, ANKRD16, MBOAT7, C3orf18*, etc. *ADK*, an enzyme protein-coding gene, encodes adenosine kinase to participate in adenosine metabolism. Previous studies have indicated that *ADK* may work as an epigenetic regulator of neurogenic genes, which is important for neuronal proliferation and plasticity, and is involved in developmental processes of the cerebrum and cerebellum [38]. In addition, *ADK* expression changes in the adult brain are also associated with various cognitive deficits. For example, increased *ADK* expression can enhance neuronal excitability, while decreased *ADK* expression is associated with epilepsy and brain injury [39]. A recent review of *ADK* summarized its epigenetic effects in neurodevelopmental disorders, brain injury, and epilepsy [40]. As a gene-encoded specific protein, *ANKRD16* (ankyrin repeat domain 16) contains ankyrin repeats, which are involved in tRNA modification and can prevent neuronal loss caused by editing defective tRNA synthase. One study found that *ANKRD16* deficiency causes extensive protein aggregation and loss of neurons [41]. Although no studies have shown a link between these genes and PTSD, they have provided new candidate genes for the pathogenesis of PTSD, and further studies are needed to confirm their association.

Additionally, in the PWAS analysis of different gender populations, multiple candidate genes associated with PTSD were identified in the male and female populations, such as *GLO1* and *GSTZ1* in females and *AKT3* and *MAPK8IP3* in males. Among the candidate genes in the female population, the *GLO1* gene was discussed in the previous passages on overlapping genes; *GSTZ1* (glutathione S-transferase Zeta 1) is a member of the glutathione S-transferase super-family, responsible for glutathione-dependent metabolism and involved in the regulation of oxidative stress [42]. Previous studies have found many genetic polymorphisms in the coding and promoter regions of the *GSTZ1* gene, having significant effects on the protein’s kinetic characteristics and/or its expression levels [43]. Another related study found that polymorphisms in the *GSTZ1* gene were associated with early-onset susceptibility to bipolar disorder [44]. Furthermore, a gene–environment interaction study suggested that *GSTZ1* is related to genes associated with harsh punitive parenting, which may contribute to social anxiety in adolescence [42]. The possible mechanism is that this gene is involved in biological pathways that regulate oxidative stress and glutamate neurotransmission associated with anxiety-like behavior and may be sensitive to stressful life events during development. Among candidate genes in the male population, *AKT3* (serine/threonine kinase 3) is highly expressed in the brain and hippocampal pyramidal cells. Substantial evidence has implicated that *AKT3* signaling is involved in cell proliferation of newborn neurons, cellular neurite outgrowth, and neurogenesis in the dentate gyrus of the hippocampus [45]. Previous studies have found that mutations of the *AKT3* gene in humans are associated with a wide spectrum of developmental disorders including extreme megalocephaly [46]. The correlation between *AKT3* and hippocampal nerve has been confirmed in animal experiments. For example, in mouse models, *AKT3*- mammalian target of rapamycin (mTOR) signaling regulates hippocampal neurogenesis in adult mice [45], and *AKT3* deficiency impairs spatial cognition and long-term potentiation of hippocampal CA1 by downregulating mTOR [47]. The studies mentioned above provided important clues for the potential biological mechanism of genes identified as affecting the development of PTSD, such as the sensitivity to stressful life events and the hippocampal nerves. In the PWAS analysis results of our study, no overlapping genes were found among the significant candidate genes in by-sex populations, suggesting that the pathogenesis of PTSD in by-sex populations may be mediated by different candidate genes and genetic factors. However, according to the current study, biological mechanisms directly associated with PTSD and gender-specific pathway/signaling changes were not found in these candidate genes. Further studies are needed to confirm our conclusions.

In the TWAS analysis, we also found multiple candidate genes significantly associated with PTSD, such as *RIMS2, CHMP1A, SIRT5, GLO1, MADD, AKAP9, PLXNA4*, etc. Previous studies have identified these genes as being involved in nervous system function or constituting risk genes for neurological diseases. Among them, *RIMS2* (regulating synaptic membrane exocytosis 2) encodes presynaptic proteins involved in synaptic membrane exocytosis and is mainly expressed in the brain. In a genome-wide survival study, *RIMS2* was identified as a novel synaptic locus and polygenic score for cognitive disease progression in Parkinson’s disease (PD) [48]. Furthermore, another study found upregulation of the *RIMS2* gene in schizophrenia, implicating altered cytomatrix active region gene expression of synapses in the amygdala for dysfunction in the pathophysiology of schizophrenia [49]. The protein encoded by *CHMP1A* (charged multivesicular body protein 1A) is an important regulator of cerebellar development. Previous studies have not only found that loss-of-function mutations in human *CHMP1A* lead to reduced cerebellar size (pontocerebellar hypoplasia) and cerebral cortex size (microcephaly) [50], but also identified *CHMP1A* as a new candidate gene for late-onset PD [51]. As for *SIRT5* (sirtuin 5), studies have shown that it is primarily expressed in brain neurons and endothelial cells and is involved in the regulation of key pathways in brain metabolism that may contribute to the adaptive response of the brain to stress and neurodegenerative processes [52]. In addition to the genes discussed above, other candidate genes revealed by TWAS analysis might be involved in the pathogenesis of PTSD at gene expression level. In addition, the TWAS analysis results for different gender populations were similar to the PWAS results; that is, no overlapping candidate genes were found in different genders, suggesting that the occurrence of PTSD in different gender populations may be associated with the pathogenesis caused by different genes.

In brain imaging analysis, we found that the *MADD* and *GLO1* genes were associated with several different brain imaging phenotypes. Among them, *MADD* was significantly associated with 19 brain imaging phenotypes, mainly involving the putamen, pallidus, left superior parietal, right cingulate gyrus, hippocampus and internal capsule brain region. *GLO1* was associated with four brain imaging phenotypes, including right bankssts area, right lateral–orbito-frontal brain regions and functional connectivity. Previous etiological studies of PTSD have identified three brain regions that may be involved in the pathophysiology of PTSD: the amygdala, medial prefrontal cortex (PFC), and hippocampus [34,53]. It is generally accepted that the PFC–hippocampal–amygdala network plays an indispensable role in mediating PTSD, and dysfunction of these brain regions is believed to underlie the development of PTSD because they are not only involved in the encoding process of threat-related stimulation, fear conditioning, explicit and emotional memory, but also have many neurochemical pathways/signals that may directly or indirectly affect PTSD [53]. The results of our brain imaging analysis showed that these two genes may be associated with several different brain regions. To some extent, this reinforces the possibility that *MADD* and *GLO1* are the candidate genes for PTSD pathogenesis, especially the *MADD* gene.

There are advantages to our study. First, our study selected the largest and latest PTSD GWAS summary data and combined them with two human brain reference proteomes to perform PWAS analysis, which can identify candidate genes associated with PTSD at the proteomic level. Second, we performed TWAS analysis to identify genes associated with PTSD at gene expression level and found overlapping genes by comparing the results of the PWAS and TWAS to improve the reliability of the results.

Limitations also should be noted in our research. First, our research mainly focused on European ancestry; when applying the research results to different populations, the impact of ethnic differences on the results should be considered. Second, to date, the human brain proteome and transcriptome reference weights used in this study are the largest data available. In the future, the data remain to be combined with larger reference weights to explore the candidate genes related to PTSD in the gene expression process. Finally, although previous studies have suggested that the genes identified in our study play a certain role in neurological function and disease, there is no direct evidence that they are associated with PTSD, and further research is needed to verify this. Despite these limitations, our research still revealed the latest results of the human brain proteome of PTSD. In future research, a more complete and larger sample size proteome analysis will undoubtedly refine the research on diseases and proteomics.

## 5. Conclusions

Our study identified multiple candidate genes associated with PTSD at the human proteome and transcriptome levels, which may act based on their respective *cis*-regulating brain protein abundance or *cis*-regulating gene expression in the development of PTSD. Moreover, we provided new clues to the pathogenesis of PTSD by identifying brain regions that may be involved in it.

## Figures and Tables

**Table 1 genes-13-01341-t001:** Proteome-wide association study (PWAS) results for PTSD.

Reference Weight	Population	CHR	SNP	Genes	Z-Score	*p*-Value
ROS/MAP	All	10	rs1908337	*ADK*	−2.47	3.00 × 10^−5^
		19	rs36655	*MBOAT7*	−2.51	2.26 × 10^−3^
	Female	2	rs2706762	*PCYOX1*	2.29	2.70 × 10^−4^
		1	rs2229540	*AKR1A1*	−2.27	1.36 × 10^−2^
		7	rs6461725	*FAM221A*	3.25	2.48 × 10^−2^
		20	rs1556876	*CSE1L*	−2.51	4.13 × 10^−2^
	Male	19	rs2335524	*TRAPPC5*	−2.84	2.60 × 10^−5^
		1	rs12404207	*BCAN*	2.80	8.14 × 10^−4^
Banner	All	3	rs13091933	*C3orf18*	3.09	4.85 × 10^−44^
		6	rs4507599	*TPMT*	3.29	2.38 × 10^−4^
		10	rs2380205	*ANKRD16*	−3.30	6.85 × 10^−3^
		16	rs11647479	*MTSS1L*	−3.13	3.79 × 10^−2^
		11	rs11570115	*MADD*	−2.13	4.9 × 10^−2^
	Female	14	rs1544707	*GSTZ1*	−2.60	1.93 × 10^−3^
		6	rs1049346	*GLO1*	2.49	4.89 × 10^−3^
		1	rs747862	*FBXO6*	−2.86	6.61 × 10^−3^
	Male	3	rs399484	*C3orf18*	2.59	7.07 × 10^−31^
		1	rs320330	*AKT3*	−2.02	6.91 × 10^−4^
		16	rs2235487	*MAPK8IP3*	−2.35	1.77 × 10^−3^
		10	rs7894083	*ANKRD16*	−3.16	3.58 × 10^−3^
		12	rs10770818	*RECQL*	2.21	9.22 × 10^−3^

Abbreviations: CHR, Chromosome; SNP, Single nucleotide polymorphism; *p*-value, Analytical permutation *p*-values; ROS/MAP, Religious Orders Study and Rush Memory and Aging Project; Banner, Banner Sun Health Research Institute.

**Table 2 genes-13-01341-t002:** Transcriptome-wide association study (TWAS) results for PTSD.

Reference Weight	Population	CHR	SNP	Genes	Z-Score	*p*-Value
Rnaseq	All	2	rs7557285	*FBXO41*	3.44	9.85 × 10^−6^
		5	rs11747154	*PCDHA9*	−2.27	5.09 × 10^−3^
		1	rs949571	*ZBTB41*	−2.11	1.54 × 10^−3^
		1	rs34001546	*C1orf54*	−3.09	4.28 × 10^−3^
		6	rs9477555	*KIF13A*	4.32	5.55 × 10^−3^
	Female	16	rs2377058	*CHMP1A*	−2.99	5.09 × 10^−4^
		5	rs6879760	*PCDHA9*	−2.69	4.17 × 10^−4^
		6	rs1049346	*GLO1*	2.50	1.41 × 10^−3^
		22	rs4474965	*DGCR8*	3.76	2.33 × 10^−3^
		1	rs34001546	*C1orf54*	−2.76	5.15 × 10^−3^
	Male	3	rs2267846	*P4HTM*	2.13	1.82 × 10^−43^
		1	rs6690515	*ATAD3A*	−4.10	2.59 × 10^−4^
		8	rs4876218	*CLN8*	3.04	5.37 × 10^−4^
		4	rs4696175	*TRIM2*	−3.13	2.87 × 10^−3^
		19	rs383547	*RAB3D*	−2.43	6.75 × 10^−3^
Splicing	All	7	rs7798233	*AKAP9*	2.51	1.51 × 10^−13^
		7	rs741664	*PLXNA4*	−3.69	1.24 × 10^−4^
		6	rs4712047	*SIRT5*	2.64	4.81 × 10^−3^
		11	rs17787912	*MADD*	−2.43	1.23 × 10^−2^
		8	rs17817071	*RIMS2*	−2.64	3.84 × 10^−2^
	Female	2	rs11681740	*COL5A2*	−2.21	1.69 × 10^−6^
		5	rs891943	*SLC6A7*	3.63	4.47 × 10^−4^
		11	rs4757650	*LDHA*	−3.20	1.50 × 10^−3^
		17	rs2010838	*AMZ2*	3.28	1.92 × 10^−3^
		5	rs7700814	*GLRX*	−3.02	2.29 × 10^−3^
	Male	7	rs13229505	*AKAP9*	2.57	1.19 × 10^−10^
		7	rs741664	*PLXNA4*	4.15	4.56 × 10^−4^
		19	rs17211813	*FKBP8*	3.45	3.23 × 10^−4^
		7	rs38417	*GGCT*	2.91	5.55 × 10^−4^
		4	rs7375984	*FSTL5*	3.53	3.52 × 10^−4^

Abbreviations: CHR, Chromosome; SNP, Single nucleotide polymorphism; *p*-value, Analytical permutation *p*-values.

**Table 3 genes-13-01341-t003:** Comparative results of the PWAS and TWAS analyses in different populations.

Population	Genes	Group	Z-Score	*p*-Value
All	*MADD*	PWAS (Banner)	−2.13	4.90 × 10^−2^
		TWAS (Splicing)	−2.43	1.23 × 10^−2^
Female	*GLO1*	PWAS (Banner)	2.49	4.89 × 10^−3^
		TWAS (Rnaseq)	3.14	1.41 × 10^−3^

Abbreviations: PWAS, Proteome-wide association study; TWAS, Transcriptome-wide association study; *p*-value: < 0.017 indicates significant association, and 0.017 < *p*-value < 0.05 indicates suggestive association.

**Table 4 genes-13-01341-t004:** Brain imaging traits related to MADD and GLO1 genes.

Gene	Brain Imaging Traits	*p*-Value
*MADD*	Mean MO in pontine crossing tract on FA skeleton	5.8 × 10^−12^
	Median T2star in left putamen	1.1 × 10^−11^
	Median T2star in right putamen	3.6 × 10^−10^
	Volume of Pallidum in the left hemisphere	8.7 × 10^−10^
	Volume of Pallidum in the right hemisphere	9.9 × 10^−10^
	Mean intensity of Accumbens area in the left hemisphere	1.2 × 10^−8^
	Volume of superior parietal in the left hemisphere	1.3 × 10^−8^
	Weighted-mean L3 in tract right cingulate gyrus part of cingulum	1.3 × 10^−8^
	Mean MD in cingulum hippocampus on FA skeleton (right)	1.3 × 10^−8^
	Mean ICVF in anterior limb of internal capsule on FA skeleton (right)	1.6 × 10^−8^
*GLO1*	Functional connectivity, connection 508 dimensionality 100	8.7 × 10^−7^
	Area of bankssts in the right hemisphere	9.3 × 10^−7^
	Functional connectivity, connection 4 dimensionality 100	2.8 × 10^−6^
	Volume of lateral–orbito-frontal in the right hemisphere	2.9 × 10^−6^

Abbreviations: Phenotype ID and brief description from Oxford Brain Imaging Genetics Server-BIG40; MO, diffusion tensor mode; FA, fractional anisotropy; MD, mean diffusivity; ICVF, intra-cellular volume fraction. *p*-value < 5 × 10^−8^ indicates the genome-wide significant association, and *p*-value < 1.27 × 10^−5^ indicates suggestive association.

## Data Availability

The datasets used and/or analyzed in the current study are available from the corresponding author on reasonable request.

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
