# Peer review of "Brain Proteome-Wide Association Study Identifies Candidate Genes that Regulate Protein Abundance Associated with Post-Traumatic Stress Disorder"

_genes, 2022, doi:10.3390/genes13081341_

Round 1
Reviewer 1 Report
Zhen Zhang, Peilin Meng et al present a PWAS and TWAS of PTSD. They examine two related GWAS datasets, and identify a number of genes of potential interest.
Although I think this is an interesting question, I have a number of methodological concerns with this manuscript as it stands, and I think it would need considerable revision.
Major concerns:
- The GWAS sets are used are the first (PGC1) and second (PGC2) GWAS from the Psychiatric Genomics Consortium working groups on PTSD. The data contained within PGC1 is entirely contained within and extended in PGC2. As this is the case, it is unclear to me what the value is of using both datasets - the authors should either focus their analyses on PGC2, or they need to clearly justify why it is useful to consider PGC1 as well.
- The descriptions of TWAS and PWAS in the introduction (lines 74-87) need considerable improvement. PWAS does not detect phenotypic associations mediated by protein function - it collapses variant-level associations together into single gene values (determined by the association of variants with protein levels), and then assesses the combined association of these collapsed variants with an outcome. This does not imply mediation of the association between the variants and the outcome by protein levels - there could be separate effects of the variants on the protein and on the outcome. The PWAS section should therefore be reworded to more accurately describe what PWAS is doing. Similarly, TWAS collapses variant-level signals together according to the association of the variants with mRNA levels - this is not well described by the existing sentence on lines 83 and 84. The statement that TWAS can eliminate most of the effects of linkage disequilibrium (line 85) is not true - linkage disequilibrium is a confounder when interpreting TWAS results (see for example PMID 30926968). This statement is also not supported by the cited paper, which is talking about the value of using gene set analysis to eliminate the bias of linkage disequilibrium from TWAS.
- The methods need considerably more detail. The quality control procedures used to generate the summary statistics for GWAS, protein levels, and RNA levels used in the paper need to be summarised (in a Supplement if not in the main paper). Details of how FUSION works need to be provided, as well as details on the specific parameters used in running FUSION in this analysis.
- It is unclear why different proteome datasets were used with different GWAS datasets. Both proteome datasets should be used with PGC2, or a strong justification for this difference needs to be given.
- The method for functional exploration of this data used is based on gene expression data. As such, it is not appropriate for use in this case, because it does not take account of biases due to the use of GWAS data to define gene-level associations (such as linkage disequilibrium). A dedicated method for this type of analysis such as e-MAGMA or JEPEGMIX2-P should be used. Even if the enrichment analysis used were appropriate, the full genome should not be used as the backbone, as many of the genes in question are not measured in the PWAS and TWAS data used.
- By-sex results are reported, but not described in the methods. How are these conducted? What is by-sex: GWAS data? Protein expression data? RNA expression data? All of the above? There needs to be a full explanation of these analyses in the methods
- When contrasting the results of PWAS and TWAS data, the authors disregard the distinction between by-sex and sex-combined analyses. This brings up issues of multiple testing - either the results of the by-sex analyses are primary (in which case they need to be treated as additional tests) or they are secondary (in which case they shouldn't be considered when contrasting the results).
Minor concerns (in order of appearance):
- Abstract: Post-traumatic stress disorder is capitalised - elsewhere it would usually be lower case.
- Abstract: The phrase "Such as such as" is jarring - "Examples include" would be preferable
- Intro, line 64: please provide details of the risk loci associated with PTSD
- Intro, line 64-66: the GxE studies cited here all predate effective GWAS of PTSD, and as such are not based on GWAS but on the candidate gene literature (the assumptions of which have not been supported by findings from GWAS). This needs to be reworded (or removed as it is only tangentially relevant to the paper).
- Intro, line 87: the example given for the value of TWAS in psychiatric disorders is not from a psychiatric disorder - it would be better to use a psychiatric example.
- Methods: multiple testing correction in the Brain imaging analysis exploration is not discussed and needs to be.
- Results: the text of the results largely reflects the data in the table - this redundancy should be removed.
- Figure 1: The term "rich factor" needs to be defined in the methods and in this figure.
- The institutional review board statement - while I agree that there is no separate IRB statement needed here, the statement given is not understandable. This statement should make clear that the original studies were conducted with appropriate IRB approvals.
Note: The discussion largely addresses results that are based on methods that I think will need to change in accordance with my comments above. As such, I have not made extensive comments on the discussion at this point.
Author Response
Response to Reviewer 1 Comments
Dear editor and reviewers,
We are grateful that Genes gave us an opportunity to reconsider our revised manuscript titled “Brain proteome-wide association study identifies candidate genes that regulate protein abundance associated with post-traumatic stress disorder” (Manuscript ID: genes-1639995). In the following, we addressed the comments from the reviewers very carefully. And the significant changes were made in the revised manuscript according to the comments.
Major concerns:
Point 1: The GWAS sets are used are the first (PGC1) and second (PGC2) GWAS from the Psychiatric Genomics Consortium working groups on PTSD. The data contained within PGC1 is entirely contained within and extended in PGC2. As this is the case, it is unclear to me what the value is of using both datasets - the authors should either focus their analyses on PGC2, or they need to clearly justify why it is useful to consider PGC1 as well.
Response 1: Thank you for your helpful comments.
The original intention of our study was to analyse PTSD using two GWAS datasets as the discovery group and the replication group, respectively. However, we have confused the fact that the PGC2 data set included the PGC1 data. According to your guidance, we have removed the description and analysis of GWAS dataset 1 (PGC1) in our research and only focused on the analysis of PTSD on GWAS dataset 2 (PGC2).
Thanks!
Point 2: The descriptions of TWAS and PWAS in the introduction (lines 74-87) need considerable improvement. PWAS does not detect phenotypic associations mediated by protein function - it collapses variant-level associations together into single gene values (determined by the association of variants with protein levels), and then assesses the combined association of these collapsed variants with an outcome. This does not imply mediation of the association between the variants and the outcome by protein levels - there could be separate effects of the variants on the protein and on the outcome. The PWAS section should therefore be reworded to more accurately describe what PWAS is doing. Similarly, TWAS collapses variant-level signals together according to the association of the variants with mRNA levels - this is not well described by the existing sentence on lines 83 and 84. The statement that TWAS can eliminate most of the effects of linkage disequilibrium (line 85) is not true - linkage disequilibrium is a confounder when interpreting TWAS results (see for example PMID 30926968). This statement is also not supported by the cited paper, which is talking about the value of using gene set analysis to eliminate the bias of linkage disequilibrium from TWAS.
Response 2: Thank you for your helpful comments.
We are sorry for the inappropriate descriptions. According to your guidance, we have made appropriate improvements in the descriptions of PWAS and TWAS. Please see the following description and line 74-95 on page 2 in the revised manuscript.
“With the continuous improvement of GWAS methods and the emergence of high-throughput proteome sequencing technology in complex tissues, GWAS not only promoted the application of proteome-wide association studies (PWAS) and TWAS, in complex human diseases. PWAS is a protein-centric approach to genetic association research, which considers the proteomic context of genetic variation and its functional effects, and decomposes variation-level associations into individual gene values (determined by the association between variation and protein level), then evaluating the combined association of these variants with outcomes[1]. What’s more, as a gene-based approach, PWAS aggregates signals from all variants affecting the same protein-coding gene which helps to detect genes with dominant and recessive effects[2].Different from GWAS, PWAS associations are supported by specific functional effects in encoding genes, making them more interpretable. For example, Thomas S. Wingo et al. identified 19 genes in the PWAS analysis that were consistent with causality in depression, acting through cis-regulated abundance of their respective brain proteins[3]. For TWAS, it was often used to identify genes whose expression is significantly associated with complex traits without directly measuring their expression levels. Specifically, TWAS used genotype and expression data from an external reference panel to determine the association between genetic variation and gene expression, then identified genes whose cis-regulated expression are associated with diseases or phenotypes by integrating gene expression data with genome-wide associations from large-scale GWAS[4]. At present, TWAS has been widely applied in the study of psychiatric disorders. For example, a TWAS study based on GWAS data identified 26 risk genes whose cis-regulated expression were significantly associated with anxiety[5].”
Thanks!
Point 3: The methods need considerably more detail. The quality control procedures used to generate the summary statistics for GWAS, protein levels, and RNA levels used in the paper need to be summarised (in a Supplement if not in the main paper). Details of how FUSION works need to be provided, as well as details on the specific parameters used in running FUSION in this analysis.
Response 3: Thank you for your helpful comments.
We are sorry for the missing description for methods. According to your helpful comment, we have added detailed description of quality control procedures and standards for GWAS summary statistics, protein levels, and mRNA levels in the corresponding methods section of our article. Among them, it is worth noting that the FUSION pipeline provides reference weights for gene expression(mRNA) that can be directly used for analysis, so we couldn’t provide quality control procedures and standards for mRNA levels. In addition, we have supplemented descriptions of FUSION operation and specific parameters in PWAS and TWAS analysis. Please see the following description and line 117-122,141-151,173-189,156-168 on page 3-4 in the revised manuscript.
- Quality control procedures:
For GWAS, “Quality control was carried out using PGC pipeline RICOPILI in accordance with sample exclusion criteria: using SNPs with call rates >95%, samples with call rates <98%, with deviation from expected inbreeding coefficient (fhet < −0.2 or >0.2), or with a sex discrepancy between reported and estimated sex based on inbreeding coefficients calculated from SNPs on X chromosomes were excluded.”
For protein levels, “Quality control criteria carried out for protein reference weight data were as follows: excluding proteins with missing value (global internal standards were used to check for proteins outside of the 95% confidence interval and set to missing); each protein abundance was scaled to sample specific total protein abundance to eliminate the effect of protein loading differences; Outlier samples were identified and removed by iterative principal component analysis (in each iteration, samples with more than four standard deviations were removed from the mean of the first or second principal component). Finally, protein abundance was normalized by residual of the linear regression to eliminate the effects of protein batch, sex, age at death, and post-mortem interval. Detailed information on proteome sequencing, quality control, and standardization can be obtained in the study by Wingo et al[6].”
For mRNA levels, “The gene expression data used in our TWAS analysis were derived from two human brain gene expression reference weights (Rnaseq and Splicing) provided by FUSION Pipeline. Both datasets were obtained from the dorsolateral prefrontal cortex (dPFCS) of 452 European individuals collected by the CommonMind Consortium (CMC), and detailed data information is available in the FUSION pipeline.”
- Details of FUSION operation and parameters during PWAS analysis:
“Using the FUSION pipeline (http://gusevlab.org/projects/fusion/ ), we integrated PTSD GWAS dataset with two different reference human brain proteomes(ROS/MAP[7] and Banner[8])to perform the PWAS analysis[4]. Specifically, in the FUSION pipeline, the protein weights of two human brain proteomes, ROS/MAP and Banner, were first estimated respectively (the two analysis processes were consistent); then calculated PTSD genetic effect (GWAS z-score), and combined with pre-calculated brain proteome reference weight (z-score × proteome weight) to evaluate the effects of significant SNP in GWAS on the protein abundance. Finally, FUSION identified candidate genes associated with PTSD regulating the abundance of proteins in the brain. The default settings and parameters recommended by FUSION were used in the analysis. (The main operating parameters are as follows: --sumstats, --weights, --weights_dir, --ref_ld_chr, --chr, --out, --perm, --GWASN; Specific parameter information can be queried on the FUSION pipeline)”
- Details of FUSION operation and parameters during TWAS analysis:
“TWAS analysis was also conducted based on FUSION pipeline, which were similar to PWAS analysis in principle and parameters. Different from PWAS analysis, the external expression reference panel used for TWAS analysis was based on mRNA expression level. Briefly, in FUSION pipeline, we first calculated the reference weights of Rnaseq and Splicing human brain gene expression panels through the multiple prediction models running in FUSION, and selected the best reference weights; then calculated PTSD genetic effects (TWAS z-score), and combined with gene expression weights (z-score ×weight) for TWAS analysis to evaluate association between gene expression levels and PTSD, and to identify candidate genes (based on gene expression levels) associated with PTSD. The default settings and parameters recommended by FUSION were used in TWAS analysis.”
Thanks!
Point 4: It is unclear why different proteome datasets were used with different GWAS datasets. Both proteome datasets should be used with PGC2, or a strong justification for this difference needs to be given.
Response 4: Thank you for your helpful comments.
We agree with you! The original intention of our study was to analyse PTSD using different GWAS datasets as the discovery group and the replication group, respectively. However, we have confused the fact that the PGC2 data set included the PGC1 data. According to your guidance, we only retained one PTSD GWAS dataset, PGC2, for our study. And some changes have been made in results and discussion based on integration two human brain proteomics data (ROS/MAP and Banner) with PGC2 for PWAS analysis. Please see the following description and line 156-158 on page 4 and line 203-215 on page 5 in the revised manuscript.
1)“Using the FUSION pipeline (http://gusevlab.org/projects/fusion/ ), we integrated PTSD GWAS dataset with two different reference human brain proteomes(ROS/MAP[7] and Banner[8])to perform the PWAS analysis[4].”
2) “PWAS Analysis Results”
In PWAS analysis combined with ROS/MAP proteome reference weights, a total of 8 candidate genes associated with PTSD were identified, such as ADK (PPWAS-ROS/MAP= 3.00×10-5) in all population; PCYOX1 (PPWAS-ROS/MAP= 2.70×10-4) in females; and BCAN (PPWAS-ROS/MAP= 8.14×10-4) in males; et al.
Another PWAS analysis combined with Banner proteome reference weights identified 13 candidate genes significantly associated with PTSD, such as MADD (PPWAS-Banner= 4.9×10-2) in all population; GLO1 (PPWAS-Banner= 4.89×10-3) in females; and C3orf18 (PPWAS-Banner= 7.07×10-31) in males; et al. Notably, C3orf18(PPWAS-all= 4.85×10-44; PPWAS-male= 7.07×10-31) and ANKRD16(PPWAS-all= 6.85×10-3; PPWAS-male= 3.58×10-3) overlapped in all population and male population in Banner proteome reference weight group. Table 1 summarizes the detailed results of the PWAS analysis.
Table 1. Proteome-wide association studies (PWAS) results of PTSD.
|
Reference weight |
Population |
CHR |
SNP |
Genes |
Z-score |
P-value |
|
ROS/MAP |
All |
10 |
rs1908337 |
ADK |
-2.47 |
3.00×10-5 |
|
19 |
rs36655 |
MBOAT7 |
-2.51 |
2.26×10-3 |
||
|
Female |
2 |
rs2706762 |
PCYOX1 |
2.29 |
2.70×10-4 |
|
|
1 |
rs2229540 |
AKR1A1 |
-2.27 |
1.36×10-2 |
||
|
7 |
rs6461725 |
FAM221A |
3.25 |
2.48×10-2 |
||
|
20 |
rs1556876 |
CSE1L |
-2.51 |
4.13×10-2 |
||
|
Male |
19 |
rs2335524 |
TRAPPC5 |
-2.84 |
2.60×10-5 |
|
|
1 |
rs12404207 |
BCAN |
2.80 |
8.14×10-4 |
||
|
Banner |
All |
3 |
rs13091933 |
C3orf18 |
3.09 |
4.85×10-44 |
|
6 |
rs4507599 |
TPMT |
3.29 |
2.38×10-4 |
||
|
10 |
rs2380205 |
ANKRD16 |
-3.30 |
6.85×10-3 |
||
|
16 |
rs11647479 |
MTSS1L |
-3.13 |
3.79×10-2 |
||
|
11 |
rs11570115 |
MADD |
-2.13 |
4.9×10-2 |
||
|
Female |
14 |
rs1544707 |
GSTZ1 |
-2.60 |
1.93×10-3 |
|
|
6 |
rs1049346 |
GLO1 |
2.49 |
4.89×10-3 |
||
|
1 |
rs747862 |
FBXO6 |
-2.86 |
6.61×10-3 |
||
|
Male |
3 |
rs399484 |
C3orf18 |
2.59 |
7.07×10-31 |
|
|
1 |
rs320330 |
AKT3 |
-2.02 |
6.91×10-4 |
||
|
16 |
rs2235487 |
MAPK8IP3 |
-2.35 |
1.77×10-3 |
||
|
10 |
rs7894083 |
ANKRD16 |
-3.16 |
3.58×10-3 |
||
|
12 |
rs10770818 |
RECQL |
2.21 |
9.22×10-3 |
Abbreviations:CHR, Chromosome; SNP, Single nucleotide polymorphism; P-value, Analytical permutation P-values.
Thanks!
Point 5: The method for functional exploration of this data used is based on gene expression data. As such, it is not appropriate for use in this case, because it does not take account of biases due to the use of GWAS data to define gene-level associations (such as linkage disequilibrium). A dedicated method for this type of analysis such as e-MAGMA or JEPEGMIX2-P should be used. Even if the enrichment analysis used were appropriate, the full genome should not be used as the backbone, as many of the genes in question are not measured in the PWAS and TWAS data used.
Response 5: Thank you for your helpful comments.
We are sorry for the inappropriate description. We agree with you. When we used Metascape's online analysis tool for enrichment analysis, we did not fully consider the bias caused by using GWAS data to define gene-level association, resulting in errors in the enrichment analysis results. Thus, in consideration of the deadline and workload of revision, we have decided to delete the incorrect enrichment analysis method and results from our research.
Thanks!
Point 6: By-sex results are reported, but not described in the methods. How are these conducted? What is by-sex: GWAS data? Protein expression data? RNA expression data? All of the above? There needs to be a full explanation of these analyses in the methods.
Response 6: Thank you for your helpful comments.
We are very sorry for the confused places. The by-sex analysis data in our study were derived from summary data of different genders in PTSD GWAS. We obtained significant independent SNPs data of PTSD in total population,males and females from the GWAS dataset, and combined them with different proteome reference weights and gene expression reference weights to conduct PWAS and TWAS analysis in total population , males and females, respectively. According to your helpful suggestion, we have added descriptions of by-sex data in methods section that introduces the GWAS dataset. Please see the following description and line 127-131 on page 3 in the revised manuscript.
“It is worth noting that PGC-PTSD not only carried out GWAS analysis of PTSD in total population, but also performed GWAS summary statistics on PTSD in by-sex population. We only selected data from European ancestry including 1,161,480 significantly independent SNPs in the total population, and 1,155,985 and 1,169,775 significantly independent SNPs in the male and female populations, respectively.”
Thanks!
Point 7: When contrasting the results of PWAS and TWAS data, the authors disregard the distinction between by-sex and sex-combined analyses. This brings up issues of multiple testing - either the results of the by-sex analyses are primary (in which case they need to be treated as additional tests) or they are secondary (in which case they shouldn't be considered when contrasting the results).
Response 7: Thank you for your helpful comments.
We are very sorry for the confused places. The by-sex analysis data in our article were derived from summary data of by-sex PTSD GWAS,and we compared the results of PWAS and TWAS in different populations. According to your guidance, and considering the impact of multiple testing on different population groups, we adopted the corrected P-value of multiple testing when contrasting the results. Please see the following description and line 230-238 on page 6-7 in the revised manuscript.
“After comparing the results of PWAS and TWAS analysis in different populations, we found an overlapping gene MADD (PPWAS-Banner = 4.90×10-2, PTWAS-Splicing = 1.23×10-2) in all population and an overlapping gene GLO1(PPWAS-Banner = 4.89×10-3, PTWAS-Rnaseq = 1.41×10-3) in female population, but no overlapping gene was detected in male population. Details were shown in Table 3.
Considering the influence of different population groups, we used the permutation-based P-value (0.05/3 =0.017) corrected by multiple testing as the threshold in the comparison results. P-value < 0.017 represents significant association, and 0.017 < P-value < 0.05 represents suggestive association.”
Table 3. Comparative results of PWAS and TWAS analysis in different populations
|
Population |
Genes |
Group |
Z-score |
P-value |
|
All |
MADD |
PWAS (Banner) |
-2.13 |
4.90×10-2 |
|
TWAS (Splicing) |
-2.43 |
1.23×10-2 |
||
|
Female |
GLO1 |
PWAS (Banner) |
2.49 |
4.89×10-3 |
|
TWAS (Rnaseq) |
3.14 |
1.41×10-3 |
Abbreviations: PWAS, Proteome-wide association studies; TWAS, Transcriptome-wide association studies; P-value: P-value < 0.017 represents significant association, and 0.017 < P-value < 0.05 represents suggestive association.
Thanks!
Minor concerns (in order of appearance):
Point 1: Abstract: Post-traumatic stress disorder is capitalised - elsewhere it would usually be lower case.
Response 1: Thank you for your helpful comments.
Per your guidance, we have replaced the incorrect capitalization of "Post-traumatic stress disorder" in the article with lowercase "post-traumatic stress disorder".
Thank!
Point 2: Abstract: The phrase "Such as such as" is jarring - "Examples include" would be preferable.
Response 2: Thank you for your helpful comments.
Per your guidance, we have replaced the inappropriate word "Such as such as" with "Examples include".
Thank!
Point 3: Intro, line 64: please provide details of the risk loci associated with PTSD.
Response 3: Thank you for your helpful comments.
We are sorry for the unclear description. According to your helpful comment, we have added descriptions of the genetic risk loci associated with PTSD. Please see the following description and line 63-69 on page 2 in the revised manuscript.
“Numerous genome-wide association studies (GWAS) of PTSD have revealed multiple genetic risk loci that are significantly associated with PTSD[9,10]. For example, NLGN1 gene is localized in excitatory synapses and plays an important role in learning and memory; knockout of NLGN1 can lead to loss of fear memory storage, and variation in NLGN1 may predispose individuals to suffering from higher levels anxiety and fear, potentially increasing their risk to develop PTSD after a traumatic event[9].”
Thanks!
Point 4: Intro, line 64-66: the GxE studies cited here all predate effective GWAS of PTSD, and as such are not based on GWAS but on the candidate gene literature (the assumptions of which have not been supported by findings from GWAS). This needs to be reworded (or removed as it is only tangentially relevant to the paper).
Response 4: Thank you for your helpful comments.
We are sorry for the incorrect description. According to your helpful suggestion, we deleted this sentence with wrong description.
Thank!
Point 5: Intro, line 87: the example given for the value of TWAS in psychiatric disorders is not from a psychiatric disorder - it would be better to use a psychiatric example.
Response 5: Thank you for your helpful comments.
Per your guidance, we deleted the example originally quoted and added the example of TWAS in psychiatric disorders research. Please see the following description and line 94-95 on page 2 in the revised manuscript.
“For example, a TWAS study identified 26 risk genes whose cis-regulated expression were significantly associated with anxiety.[5]”
Thanks!
Point 6: Methods: multiple testing correction in the Brain imaging analysis exploration is not discussed and needs to be.
Response 6: Thank you for your helpful comments.
We are sorry for the missing description about definition of P-value association threshold in the method section of the Brain imaging analysis exploration. We quoted to the original literature of brain imaging analysis, which not only used a 1% minor allele frequency filter, but also adjusted the P-value association threshold to avoid the effect of multiple testing correction. Please see the following description and line 198-200 on page 4 in the revised manuscript.
“To avoid multiple testing correction, the analysis used a 1% minor allele frequency (MAF) filter. And with P-value < 1 × 10-7.5 indicates the genome-wide significant association, P-value < 1 × 10-5 indicates suggestive association.”
Thanks!
Point 7: Results: the text of the results largely reflects the data in the table - this redundancy should be removed.
Response 7: Thank you for your helpful comments.
According to your guidance, we have removed redundant descriptions in the results section to. Please see the results section of the revised manuscript.
Point 8: Figure 1: The term "rich factor" needs to be defined in the methods and in this figure.
Response 8: Thank you for your helpful comments.
Based on your previous comments, we have removed the incorrect enrichment analysis method from our study. Therefore, there is no need to define the term "enrichment factor" in enrichment analysis.
Thanks!
Point 9: The institutional review board statement - while I agree that there is no separate IRB statement needed here, the statement given is not understandable. This statement should make clear that the original studies were conducted with appropriate IRB approvals.
Response 9: Thank you for your helpful comments.
According to your guidance, we have revised the description of the institutional review board statement. Please see the following description.
“Institutional Review Board Statement: The original studies providing data sources in this study were all conducted with appropriate institutional review board approval.”
Thanks!
References
- Brandes, N.; Linial, N.; Linial, M. PWAS: proteome-wide association study-linking genes and phenotypes by functional variation in proteins. Genome Biol 2020, 21, 173, doi:10.1186/s13059-020-02089-x.
- Brandes, N.; Linial, N.; Linial, M. Genetic association studies of alterations in protein function expose recessive effects on cancer predisposition. Sci Rep 2021, 11, 14901, doi:10.1038/s41598-021-94252-y.
- Wingo, T.S.; Liu, Y.; Gerasimov, E.S.; Gockley, J.; Logsdon, B.A.; Duong, D.M.; Dammer, E.B.; Lori, A.; Kim, P.J.; Ressler, K.J.; et al. Brain proteome-wide association study implicates novel proteins in depression pathogenesis. Nat Neurosci 2021, doi:10.1038/s41593-021-00832-6.
- Gusev, A.; Ko, A.; Shi, H.; Bhatia, G.; Chung, W.; Penninx, B.W.; Jansen, R.; de Geus, E.J.; Boomsma, D.I.; Wright, F.A.; et al. Integrative approaches for large-scale transcriptome-wide association studies. Nat Genet 2016, 48, 245-252, doi:10.1038/ng.3506.
- Su, X.; Li, W.; Lv, L.; Li, X.; Yang, J.; Luo, X.J.; Liu, J. Transcriptome-Wide Association Study Provides Insights Into the Genetic Component of Gene Expression in Anxiety. Front Genet 2021, 12, 740134, doi:10.3389/fgene.2021.740134.
- Wingo, A.P.; Liu, Y.; Gerasimov, E.S.; Gockley, J.; Logsdon, B.A.; Duong, D.M.; Dammer, E.B.; Robins, C.; Beach, T.G.; Reiman, E.M.; et al. Integrating human brain proteomes with genome-wide association data implicates new proteins in Alzheimer's disease pathogenesis. Nat Genet 2021, 53, 143-146, doi:10.1038/s41588-020-00773-z.
- Duncan, L.E.; Ratanatharathorn, A.; Aiello, A.E.; Almli, L.M.; Amstadter, A.B.; Ashley-Koch, A.E.; Baker, D.G.; Beckham, J.C.; Bierut, L.J.; Bisson, J.; et al. Largest GWAS of PTSD (N=20 070) yields genetic overlap with schizophrenia and sex differences in heritability. Mol Psychiatry 2018, 23, 666-673, doi:10.1038/mp.2017.77.
- Nievergelt, C.M.; Maihofer, A.X.; Klengel, T.; Atkinson, E.G.; Chen, C.Y.; Choi, K.W.; Coleman, J.R.I.; Dalvie, S.; Duncan, L.E.; Gelernter, J.; et al. International meta-analysis of PTSD genome-wide association studies identifies sex- and ancestry-specific genetic risk loci. Nat Commun 2019, 10, 4558, doi:10.1038/s41467-019-12576-w.
- Kilaru, V.; Iyer, S.V.; Almli, L.M.; Stevens, J.S.; Lori, A.; Jovanovic, T.; Ely, T.D.; Bradley, B.; Binder, E.B.; Koen, N.; et al. Genome-wide gene-based analysis suggests an association between Neuroligin 1 (NLGN1) and post-traumatic stress disorder. Transl Psychiatry 2016, 6, e820, doi:10.1038/tp.2016.69.
- Almli, L.M.; Stevens, J.S.; Smith, A.K.; Kilaru, V.; Meng, Q.; Flory, J.; Abu-Amara, D.; Hammamieh, R.; Yang, R.; Mercer, K.B.; et al. A genome-wide identified risk variant for PTSD is a methylation quantitative trait locus and confers decreased cortical activation to fearful faces. Am J Med Genet B Neuropsychiatr Genet 2015, 168b, 327-336, doi:10.1002/ajmg.b.32315.

Reviewer 2 Report
In the article "Brain proteome-wide association study identifies candidate 2 genes that regulate protein abundance associated with post-3 traumatic stress disorder" Zhang et al. use publicly available transcriptome-wide association datasets and proteome-wide association datasets to assess genome-wide data from PTSD cases from two different sources. The authors state that this technique has identified new genes that may be associated with post-traumatic stress disorder. Furthermore, the expression of the identified genes in brain regions is investigated in order to gain further insights into PTSD.
The present work is an analysis of publicly available datasets. The methodology is largely inadequately explained, so that it remains unclear to the reader whether many of the questions left open at the end are based on linguistic inaccuracies or on methodological problems. For example, details of the FUSION pipeline are not given and the reader is left in the dark as to how exactly the data sets are analyzed by this.
Unfortunately, further methodology used by the authors of the article is also described only very inaccurately. In some cases, it remains unclear how the authors arrived at certain results (for example, the results shown in Figure 1). Numerous formulations are superficial, imprecise and make no sense (for example: „As the product of gene expression in the brain, brain proteins not only play an important role in maintaining the normal function of brain cells, but also may lead to neurological diseases.” – brain proteins (?) lead to neurological diseases?). Much of the discussion is disjointed naming of genes, proteins, and their postulated function. The present work does not provide any additional benefit here, so that it is to be understood scientifically as no more than a poorly written review.
If this manuscript is indeed accepted as a scientific publication, I suggest that it should undergo critical language revision for grammatical (run-on and fragment sentences), punctuation and typographical errors, which are too numerous to mention in their entirety. The authors need more diligent proofreading, when possible by a native English speaker. Gene names are conventionally always written in italics; this must be adjusted throughout the manuscript.
Author Response
Response to Reviewer 2 Comments
Dear editor and reviewers,
We are grateful that Genes gave us an opportunity to reconsider our revised manuscript titled “Brain proteome-wide association study identifies candidate genes that regulate protein abundance associated with post-traumatic stress disorder” (Manuscript ID: genes-1639995). In the following, we addressed the comments from the reviewers very carefully. And the significant changes were made in the revised manuscript according to the comments.
Comments:
In the article "Brain proteome-wide association study identifies candidate 2 genes that regulate protein abundance associated with post-3 traumatic stress disorder" Zhang et al. use publicly available transcriptome-wide association datasets and proteome-wide association datasets to assess genome-wide data from PTSD cases from two different sources. The authors state that this technique has identified new genes that may be associated with post-traumatic stress disorder. Furthermore, the expression of the identified genes in brain regions is investigated in order to gain further insights into PTSD.
The present work is an analysis of publicly available datasets. The methodology is largely inadequately explained, so that it remains unclear to the reader whether many of the questions left open at the end are based on linguistic inaccuracies or on methodological problems. For example, details of the FUSION pipeline are not given and the reader is left in the dark as to how exactly the data sets are analyzed by this.
Unfortunately, further methodology used by the authors of the article is also described only very inaccurately. In some cases, it remains unclear how the authors arrived at certain results (for example, the results shown in Figure 1). Numerous formulations are superficial, imprecise and make no sense (for example: „As the product of gene expression in the brain, brain proteins not only play an important role in maintaining the normal function of brain cells, but also may lead to neurological diseases.” – brain proteins (?) lead to neurological diseases?). Much of the discussion is disjointed naming of genes, proteins, and their postulated function. The present work does not provide any additional benefit here, so that it is to be understood scientifically as no more than a poorly written review.
If this manuscript is indeed accepted as a scientific publication, I suggest that it should undergo critical language revision for grammatical (run-on and fragment sentences), punctuation and typographical errors, which are too numerous to mention in their entirety. The authors need more diligent proofreading, when possible by a native English speaker. Gene names are conventionally always written in italics; this must be adjusted throughout the manuscript.
Response: Thank you for your helpful comment.
We are sorry for the unclear description. Our overall research approach was to integrate the PTSD GWAS dataset with protein reference weight (ROS/MAP and Banner) and transcriptome reference weight (Rnaseq and Splicing), respectively, to conduct PWAS and TWAS analysis. By combining GWAS, TWAS and PWAS datasets, we hope to identify genetic loci associated with PTSD (at the protein and transcriptomic level) in gene expression. Then, comparing the results from PWAS and TWAS analyses we found significant genes that overlapped in the two analyses. Finally, brain imaging analysis was performed to identify brain regions associated with overlapping genes, aims to provide new insights into the pathogenesis of PTSD.
[1] According to your helpful comment, we have made a more detailed description of the method section of our article, for example, the source and quality control criteria of the PTSD GWAS dataset, details of the PWAS and TWAS operation in the FUSION pipeline, etc.
[2] Limited by publicly available datasets, it is difficult for our study to draw conclusions straightforward like experimental and clinical studies. We can only collect evidence that may be related to the pathogenesis of PTSD from previous studies, and provide some potential and new clues for further PTSD research. According to your comments, in discussion section, we have deleted the inappropriate descriptions, and described the identified candidate genes/encoded proteins in detail, and intensively explored the biological mechanisms and pathways that may be involved in PTSD.
[3] We have invited a native English speaker to improve grammatical (run-on and fragment sentences), punctuation and typographical errors in our paper, and changed the gene names to italics throughout the manuscript.
We sincerely request you to reconsider our manuscript.
Thanks!

Reviewer 3 Report
In this study, the authors combine Genome-, Transcriptome-, and Proteome-wide association studies to develop a picture of the molecular changes that may be underlie genetic susceptibility to post-traumatic stress disorder. By combining these multiple association studies, the identify a few key genes associated with PTSD. The combination of all three of these association studies is in itself a novel research strategy and stands to provide more meaningful information regarding target genes due to the analysis at multiple levels throughout the gene expression process.
Major:
1) Overall, the manuscript seems to be lacking in reference citation. Many statements have 1-2 citations or none at all. However, for example, a literature search for "PWAS and psychological disorders" returned 12 manuscripts, while "TWAS and psychological disorders" returned over 100 manuscripts.
2) The authors should provide for detail related to the studies referenced for MADD and GLO1 in the discussion. Somewhat vague statements, such as "nervous system abnormalities," fail to highlight the functional impact these genes may have in PTSD. It may need to be clarified in the text whether the statements are related to the findings of other GWAS studies or from direct scientific investigation, ie. clinical or preclinical research studies.
3) Genes/transcripts mentioned throughout the manuscript should be described as to the protein or classification of protein molecule they encode (i.e. genes encoding neurotransmitter receptors, etc). Without prior knowledge of the specific genes mentioned, the reader has to search for what each gene may encode to make sense of the significance of its identification.
4) It is somewhat unclear what data was used for the GO enrichment analysis. Is this the enrichment related to just MADD and GLO1? This should be clarified in the text of the manuscript and there should be more of a description of the analysis in the Figure 1 legend.
5) The authors identified different loci associated with PTSD in males and females. The potential role of these genes is discussed, however, is there any notable significance between these genes function in males vs. females? Do the genes suggest different pathway/signaling changes in males vs. females? This is an area that could be expanded upon in the discussion.
Minor:
There are several typographical and grammatical errors throughout the manuscript and I would suggest that the authors carefully re-read their manuscript to correct these issues. A few of these are outlined below:
1) Line 20: "...2, respectively. Such as such as EIF2B3..." should be "...2, respectively, such as EIF2B3..."
2) Line 45: "(PFC8)" should probably be "(PFC)"?
3) Line 57: "...genetic studies have been provided..." should be "...genetic studies have provided..."
4) Line 67: "studies" should be "study"
5) Line 70: "rare" should be "rarely"
6) Lines 72-74: This sentence is confusing and needs reworded.
7) Lines 88-95: These last few sentences are fragmented and disconnected making the point unclear. The authors describe their hypothesis in the final sentence of this paragraph and I think the authors are trying to make the point that disease may occur due to pathological changes at any step of the gene expression process, from transcription to translation to protein function, which may lead back to issues with transcription through negative regulation; by combining GWAS, TWAS, and PWAS data the authors are attempting to identify gene loci that are affected in PTSD at all of these steps.
8) CHR should be defined in the supplementary tables. I'm assuming this is for "chromosome."
Author Response
Response to Reviewer 3 Comments
Dear editor and reviewers,
We are grateful that Genes gave us an opportunity to reconsider our revised manuscript titled “Brain proteome-wide association study identifies candidate genes that regulate protein abundance associated with post-traumatic stress disorder” (Manuscript ID: genes-1639995). In the following, we addressed the comments from the reviewers very carefully. And the significant changes were made in the revised manuscript according to the comments.
Point 1: Overall, the manuscript seems to be lacking in reference citation. Many statements have 1-2 citations or none at all. However, for example, a literature search for "PWAS and psychological disorders" returned 12 manuscripts, while "TWAS and psychological disorders" returned over 100 manuscripts.
Response 1: Thank you for your helpful comments.
We agree with you. According to your guidance, we modified the references repeatedly quoted in our article, and quoted the latest study results of PWAS and TWAS at the present stage, as well as the most relevant citations for PTSD, to support our article and viewpoints. Please see the latest revised manuscript.
Thanks!
Point 2: The authors should provide for detail related to the studies referenced for MADD and GLO1 in the discussion. Somewhat vague statements, such as "nervous system abnormalities," fail to highlight the functional impact these genes may have in PTSD. It may need to be clarified in the text whether the statements are related to the findings of other GWAS studies or from direct scientific investigation, ie. clinical or preclinical research studies.
Response 2: Thank you for your helpful comments.
We are sorry for the unclear description. According to your helpful comment, we re-discussed MADD and GLO1 genes, cited more convincing evidence, and tried our best to find and elaborate the possible functions and pathways of these two genes affecting the pathogenesis of PTSD. Please see the following description and line 274-310 on page 8 in the revised manuscript.
“For the overlapping genes MADD and GLO1 detected in PWAS and TWAS analysis, previous studies have found that they are not only associated with neurological diseases, but also have a certain impact on neurological functions. Specifically, MADD (MAP kinase activating death domain), as an enzyme protein-coding gene, encodes a mitogen-activated protein kinase (MAPK)-activating death domain protein, which is mainly expressed in brain tissue and has various cellular functions. Such as vesicle trafficking, tumor necrosis factor-α (TNF-α) induced signal transduction and prevention of cell apoptosis[1]. Pauline E et al. have revealed the role of MADD gene in regulating physiological functions of sensory and autonomic nervous system; and deletion and mutation of MADD gene can lead to neurodevelopmental disorders, intelligence and language disorders by affecting TNF-α dependent signaling pathways and vesicular trafficking[1]. Jun Miyoshi et al. not only validated the function of MADD in the nervous system, but also found that MADD is associated with Alzheimer's Disease (AD)[2]. Another study proved that the down-regulation of MADD is associated with neuronal cell death in the brain and hippocampal neurons of AD[3]. What's more, an integrative analysis based on insomnia GWAS dataset and brain region related enhancer maps showed that MADD gene was significantly associated with insomnia, revealing the role of MADD in insomnia and other mental disorders[4]. Among these confirmed neurological functions associated with MADD, the fact that MADD can cause abnormalities in the hippocampus and neurons which is thought to underlie the development of PTSD, further enhancing the possibility that MADD is related to the pathogenesis of PTSD[5]. GLO1, as an enzyme protein-coding gene, encodes glyoxalase I and involved in detoxification of methylglyoxal(MG) in the cellular glycolysis pathway[6]. Previous studies have found that GLO1 expression and MG accumulation in the brain are associated with the pathogenesis of psychiatric disorders, such as anxiety disorder, depression, autism and schizophrenia[7]. A recent hospital-based case-control study found significant differences in the distribution of GLO1 variation, RNA expression, and enzyme activity between schizophrenia patients and controls, and suggested GLO1 is associated with dysfunction in the left middle frontal gyrus in schizophrenia[8]. In addition, animal experiments have also found that GLO1 expression is associated with anxiety-like behaviour in mice and multiple psychiatric diseases in humans[9]. However, a link between GLO1 and PTSD hasn’t been directly demonstrated. Based on existing studies, we speculated that GLO1 may be involved in the pathogenesis of PTSD by affecting the brain regions associated with PTSD. Further studies are needed to explore the association between GLO1 and PTSD. The overlap between PWAS and TWAS results highlights the consistency at protein and gene expression levels, and validates the possibility that overlapping genes are associated with the pathogenesis of PTSD from different perspectives.”
Thanks!
Point 3: Genes/transcripts mentioned throughout the manuscript should be described as to the protein or classification of protein molecule they encode (i.e. genes encoding neurotransmitter receptors, etc). Without prior knowledge of the specific genes mentioned, the reader has to search for what each gene may encode to make sense of the significance of its identification.
Response 3: Thank you for your helpful comments.
We are sorry for the unclear description. According to your helpful comment, we have provided a detailed description of the classification of proteins or protein molecules encoded by genes identified in PWAS and TWAS analysis in the discussion section. Please see lines 274-385 on pages 8-10 of the discussion section in the revised manuscript.
Thanks!
Point 4: It is somewhat unclear what data was used for the GO enrichment analysis. Is this the enrichment related to just MADD and GLO1? This should be clarified in the text of the manuscript and there should be more of a description of the analysis in the Figure 1 legend.
Response 4: Thank you for your helpful comments.
We are sorry for the inappropriate description. We performed enrichment analysis in Metascape's online analysis tool for genes identified in PWAS and TWAS analyses that were significantly associated with PTSD. However, when we used Metascape's online analysis tool for enrichment analysis, we ignored the bias caused by using GWAS data to define gene-level associations, resulting in errors in the enrichment analysis results. Considering the deadline and workload of the revision, we decided to remove the incorrect enrichment analysis methods and results from our study. Therefore, we will not expand the description of the methods and terms related to enrichment analysis in the text.
Thanks!
Point 5: The authors identified different loci associated with PTSD in males and females. The potential role of these genes is discussed, however, is there any notable significance between these genes function in males vs. females? Do the genes suggest different pathway/signaling changes in males vs. females? This is an area that could be expanded upon in the discussion.
Response 5: Thank you for your helpful comments.
We are very sorry for the confused places. According to your enlightening comments, we searched the significant genes in by-sex populations to detect their involved functions, pathways and biological pathways. Unfortunately, based on current research, no sex-specific functional, pathway/signaling changes have been found in these genes. Therefore, we re-discussed the analysis of PWAS and TWAS in different gender populations, and weakened the description of gender differences in PTSD. Please see the following description and line 328-361 on page 9 in the revised manuscript.
“Additionally, in the PWAS analysis of different gender population, multiple candidate genes associated with PTSD were identified in male and female population, respectively. Such as GLO1 and GSTZ1 in females; AKT3 and MAPK8IP3 in males. For candidate genes in the female population, the GLO1 gene has been discussed in the previous overlapping genes; GSTZ1 (glutathione S-transferase Zeta 1) is a member of the glutathione S-transferase super-family, responsible for glutathione-dependent metabolism and involved in the regulation of oxidative stress[10]. Previous studies have found a large number of genetic polymorphisms in the coding region and promoter region of GSTZ1 gene, which have significant effects on the protein's kinetic characteristics and/or its expression levels[11]. Another related study found that polymorphisms in the GSTZ1 gene were associated with early-onset susceptibility to bipolar disorder[12]. Furthermore, a gene–environment interaction study suggested that GSTZ1 is related to genes with harsh punitive parenting, which may contribute to social anxiety in adolescence[10]. The possible mechanism is that this gene is involved in biological pathways that regulating oxidative stress and glutamate neurotransmission associated with anxiety-like behavior, and may be sensitive to stressful life events during development. For candidate genes in male population, AKT3 (serine/threonine kinase 3) is highly expressed in the brain and hippocampal pyramidal cells. Substantial evidence implicated that AKT3 signaling is involved in cell proliferation of newborn neurons, cellular neurite outgrowth, and neurogenesis in the dentate gyrus of the hippocampus[13]. Previous studies have found that AKT3 gene mutations in humans are associated with a wide spectrum of developmental disorders including extreme megalencephaly[14]. The correlation between AKT3 and hippocampal nerve has been confirmed in animal experiments. For example, in mouse models, AKT3- mammalian target of rapamycin (mTOR) signaling regulates hippocampal neurogenesis in adult mice[13], and AKT3 deficiency impairs spatial cognition and long-term potentiation of hippocampal CA1 by down-regulating mTOR[15]. These studies mentioned above provide important clues for the potential biological mechanism of identified genes affecting the development of PTSD, such as the sensitivity to stressful life events and hippocampal nerves. In PWAS analysis results of our study, no overlapping genes were found among the significant candidate genes in by-sex populations, suggesting that the pathogenesis of PTSD in by-sex populations may be mediated by different candidate genes and genetic factors. However, according to the current study, biological mechanism directly associated with PTSD and gender-specific pathway/signaling changes weren’t found in these candidate genes. Further studies are needed to confirm our conclusions.”
Thanks!
Minor:
There are several typographical and grammatical errors throughout the manuscript and I would suggest that the authors carefully re-read their manuscript to correct these issues. A few of these are outlined below:
- Line 20: "...2, respectively. Such as such as EIF2B3..." should be "...2, respectively, such as EIF2B3..."
- Line 45: "(PFC8)" should probably be "(PFC)"?
- Line 57: "...genetic studies have been provided..." should be "...genetic studies have provided..."
- Line 67: "studies" should be "study"
- Line 70: "rare" should be "rarely"
6) Lines 72-74: This sentence is confusing and needs reworded.
Response 1-6): Thank you for your helpful comments.
We should apologize for our carelessness and poor English grammar. Per your guidance, we have revised these errors and carefully check our article to correct typographical and grammatical errors elsewhere in the article.
Thanks!
7) Lines 88-95: These last few sentences are fragmented and disconnected making the point unclear. The authors describe their hypothesis in the final sentence of this paragraph and I think the authors are trying to make the point that disease may occur due to pathological changes at any step of the gene expression process, from transcription to translation to protein function, which may lead back to issues with transcription through negative regulation; by combining GWAS, TWAS, and PWAS data the authors are attempting to identify gene loci that are affected in PTSD at all of these steps.
Response 7): Thank you for your helpful comments.
We are sorry for the unclear description. According to your helpful comment, we have modified the last sentences in this paragraph to better express the point of view of the article. Please see the following description and line 95-103 on page 2-3 in the revised manuscript.
“Although a large proportion of variation in complex human traits is caused by genetic factors, the mechanisms between genetic factors and traits are generally difficult to understand. The reason is that complex traits or diseases can result from pathological changes at any step in the gene expression process (from transcription to translation to protein function, or reverse transcription). Previous studies have found that many genetic factors can affect complex traits or diseases by modulating gene expression, finally altering protein abundance levels[16]. Therefore, we hypothesized that specific genetic variations affect the pathogenesis of PTSD by altering gene expression levels of proteins and transcriptomes.”
Thanks!
8) CHR should be defined in the supplementary tables. I'm assuming this is for "chromosome."
Response 8): Thank you for your helpful comments.
We are sorry for the missing definition of CHR in the supplementary tables, per your guidance, we defined the CHR in the supplementary table, that is, "Abbreviation: CHR, chromosome".
Thanks!
Reference:
- Schneeberger, P.E.; Kortum, F.; Korenke, G.C.; Alawi, M.; Santer, R.; Woidy, M.; Buhas, D.; Fox, S.; Juusola, J.; Alfadhel, M.; et al. Biallelic MADD variants cause a phenotypic spectrum ranging from developmental delay to a multisystem disorder. Brain 2020, 143, 2437-2453, doi:10.1093/brain/awaa204.
- Miyoshi, J.; Takai, Y. Dual role of DENN/MADD (Rab3GEP) in neurotransmission and neuroprotection. Trends Mol Med 2004, 10, 476-480, doi:10.1016/j.molmed.2004.08.002.
- Del Villar, K.; Miller, C.A. Down-regulation of DENN/MADD, a TNF receptor binding protein, correlates with neuronal cell death in Alzheimer's disease brain and hippocampal neurons. Proc Natl Acad Sci U S A 2004, 101, 4210-4215, doi:10.1073/pnas.0307349101.
- Ding, M.; Li, P.; Wen, Y.; Zhao, Y.; Cheng, B.; Zhang, L.; Ma, M.; Cheng, S.; Liu, L.; Du, Y.; et al. Integrative analysis of genome-wide association study and brain region related enhancer maps identifies biological pathways for insomnia. Prog Neuropsychopharmacol Biol Psychiatry 2018, 86, 180-185, doi:10.1016/j.pnpbp.2018.05.026.
- Nutt, D.J.; Malizia, A.L. Structural and functional brain changes in posttraumatic stress disorder. J Clin Psychiatry 2004, 65 Suppl 1, 11-17.
- Tian, X.; Wang, Y.; Ding, X.; Cheng, W. High expression of GLO1 indicates unfavorable clinical outcomes in glioma patients. J Neurosurg Sci 2019, doi:10.23736/S0390-5616.19.04805-7.
- Toriumi, K.; Miyashita, M.; Suzuki, K.; Tabata, K.; Horiuchi, Y.; Ishida, H.; Itokawa, M.; Arai, M. Role of glyoxalase 1 in methylglyoxal detoxification-the broad player of psychiatric disorders. Redox Biol 2022, 49, 102222, doi:10.1016/j.redox.2021.102222.
- Yin, J.; Ma, G.; Luo, S.; Luo, X.; He, B.; Liang, C.; Zuo, X.; Xu, X.; Chen, Q.; Xiong, S.; et al. Glyoxalase 1 Confers Susceptibility to Schizophrenia: From Genetic Variants to Phenotypes of Neural Function. Front Mol Neurosci 2021, 14, 739526, doi:10.3389/fnmol.2021.739526.
- Williams, R.t.; Lim, J.E.; Harr, B.; Wing, C.; Walters, R.; Distler, M.G.; Teschke, M.; Wu, C.; Wiltshire, T.; Su, A.I.; et al. A common and unstable copy number variant is associated with differences in Glo1 expression and anxiety-like behavior. PLoS One 2009, 4, e4649, doi:10.1371/journal.pone.0004649.
- Chubar, V.; Van Leeuwen, K.; Bijttebier, P.; Van Assche, E.; Bosmans, G.; Van den Noortgate, W.; van Winkel, R.; Goossens, L.; Claes, S. Gene-environment interaction: New insights into perceived parenting and social anxiety among adolescents. Eur Psychiatry 2020, 63, e64, doi:10.1192/j.eurpsy.2020.62.
- Blackburn, A.C.; Tzeng, H.F.; Anders, M.W.; Board, P.G. Discovery of a functional polymorphism in human glutathione transferase zeta by expressed sequence tag database analysis. Pharmacogenetics 2000, 10, 49-57, doi:10.1097/00008571-200002000-00007.
- Rezaei, Z.; Saadat, I.; Saadat, M. Association between three genetic polymorphisms of glutathione S-transferase Z1 (GSTZ1) and susceptibility to bipolar disorder. Psychiatry Res 2012, 198, 166-168, doi:10.1016/j.psychres.2011.09.002.
- Zhang, T.; Ding, H.; Wang, Y.; Yuan, Z.; Zhang, Y.; Chen, G.; Xu, Y.; Chen, L. Akt3-mTOR regulates hippocampal neurogenesis in adult mouse. J Neurochem 2021, 159, 498-511, doi:10.1111/jnc.15441.
- Alcantara, D.; Timms, A.E.; Gripp, K.; Baker, L.; Park, K.; Collins, S.; Cheng, C.; Stewart, F.; Mehta, S.G.; Saggar, A.; et al. Mutations of AKT3 are associated with a wide spectrum of developmental disorders including extreme megalencephaly. Brain 2017, 140, 2610-2622, doi:10.1093/brain/awx203.
- Zhang, T.; Shi, Z.; Wang, Y.; Wang, L.; Zhang, B.; Chen, G.; Wan, Q.; Chen, L. Akt3 deletion in mice impairs spatial cognition and hippocampal CA1 long long-term potentiation through downregulation of mTOR. Acta Physiol (Oxf) 2019, 225, e13167, doi:10.1111/apha.13167.
- Lappalainen, T.; Sammeth, M.; Friedländer, M.R.; t Hoen, P.A.; Monlong, J.; Rivas, M.A.; Gonzàlez-Porta, M.; Kurbatova, N.; Griebel, T.; Ferreira, P.G.; et al. Transcriptome and genome sequencing uncovers functional variation in humans. Nature 2013, 501, 506-511, doi:10.1038/nature12531.

Reviewer 4 Report
I thought the study contributed new insights into the pathogenesis of PTSD by identifying several candidate genes associated with PTSD and several brain regions that may be implicated in the pathogenesis of PTSD. I felt that the newly submitted version was an improvement over the previous one.
Author Response
Response to Reviewer 4 Comments
Dear editor and reviewers,
We are grateful that you have given us an opportunity to reconsider our revised manuscript entitled “Brain proteome-wide association study identifies candidate genes that regulate protein abundance associated with post-traumatic stress disorder” (Manuscript ID: genes-1639995). We have addressed the comments from the reviewers very carefully. In the following, we detail our responses to the comments and made the significant changes in the revised manuscript. For ease of reviewing, any significant changes in the revised manuscript are flagged in MS Word's “Track Changes” function.
Reviewer Comments and Suggestions for Authors:
I thought the study contributed new insights into the pathogenesis of PTSD by identifying several candidate genes associated with PTSD and several brain regions that may be implicated in the pathogenesis of PTSD. I felt that the newly submitted version was an improvement over the previous one.
Response:
We appreciate your previous helpful suggestions for this article, making the new submitted version a great improvement over the previous one. Based on your previous suggestions, we have carefully revised the article to better describe our research and clearly present the results. In this revision, we have made some revisions to potentially minor problems in our article according to the suggestions from other reviewers. Please see the latest revised manuscript.
Thank you again for your valuable comments on the improvement of our article!
Round 2
Reviewer 1 Report
The authors have largely addressed my previous concerns or removed the analyses to which they referred. I have two remaining concerns I note below. I have also reviewed the discussion, as I did not so previously, and do not see any major issues with it.
1. Multiple testing adjustment in the Brain imaging analysis remains unclear. First, numbers written in scientific notation are typically of the form X.XX x 10integer - 1 x 10-7.5 should be written as 3.16 x 10-8. Second, it is unclear where the thresholds for genome-wide significance and suggestive significance have come from. The authors need to explain how they reached these values.
2. TWAS and PWAS descriptions - the authors have largely addressed my previous comment, and I find both sections to be acceptably accurate descriptions of these methods. However, I still find the descriptions of PWAS is unclear. Particularly, the term "variation level associations" and the later "association between variation and protein levels" would be better phrased as "variant level associations" and "association between variants and protein levels".
Author Response
Response to Reviewer 1 Comments
Dear editor and reviewers,
We are grateful that you have given us an opportunity to reconsider our revised manuscript entitled “Brain proteome-wide association study identifies candidate genes that regulate protein abundance associated with post-traumatic stress disorder” (Manuscript ID: genes-1639995). We have addressed the comments from the reviewers very carefully. In the following, we detailed our responses to the comments and made the significant changes in the revised manuscript. For ease of reviewing, any significant changes in the revised manuscript are flagged in MS Word's “Track Changes” function.
Reviewer Comments and Suggestions for Authors:
The authors have largely addressed my previous concerns or removed the analyses to which they referred. I have two remaining concerns I note below. I have also reviewed the discussion, as I did not so previously, and do not see any major issues with it.
Major concerns:
Point 1: Multiple testing adjustment in the Brain imaging analysis remains unclear. First, numbers written in scientific notation are typically of the form X.XX x 10integer - 1 x 10-7.5 should be written as 3.16 x 10-8. Second, it is unclear where the thresholds for genome-wide significance and suggestive significance have come from. The authors need to explain how they reached these values.
Response 1: Thank you for your helpful comments.
We are very sorry for the confused places. Based on your comments, we have corrected the numbers written in scientific notation and redescribed the multiple testing correction in the Brain imaging analysis to make clear the thresholds for genome-wide significance and suggestive significance. Please see the following description and line 225-229 on page 5 in the revised manuscript.
“The analysis used minor allele frequency (MAF) > 0.01 as filter. To avoid multiple testing correction, we used multiple testing corrected P-values (0.05/3935 =1.27 × 10-5) as suggestive association threshold; to be conservative, we additionally adopted a more stringent threshold, the genome-wide significance threshold (P= 5 × 10−8), as the significant association threshold.”
Thanks!
Point 2: TWAS and PWAS descriptions - the authors have largely addressed my previous comment, and I find both sections to be acceptably accurate descriptions of these methods. However, I still find the descriptions of PWAS is unclear. Particularly, the term "variation level associations" and the later "association between variation and protein levels" would be better phrased as "variant level associations" and "association between variants and protein levels".
Response 2: Thank you for your helpful comments.
We are sorry for the inappropriate descriptions. According to your guidance, we have tried our best to provide a clear and accurate description for PWAS in the article and carefully checked our article to avoid the same mistakes elsewhere. Please see the revised manuscript.
Thanks!

Reviewer 2 Report
In revising their article, the authors could not at all do justice to the criticisms made by me and the other reviewers. I recommend that the authors fundamentally rewrite the article and make a deep linguistic revision of the text.
In many instances it remains completely unclear what is meant by the authors (e.g. p2 l90 "What's more, as a gene-based approach, PWAS aggregates signals from all variants affecting the same protein-coding gene which helps to detect genes with dominant and recessive effects") This sentence does not make any sense to me even after reading it several times.
Parts of the text even seem to be missing (e.g. p5 l341 "In PWAS analysis combined with ROS/MAP proteome reference weights, a total of 341 8 candidate genes associated with PTSD were identified, such as ADK (PPWAS-ROS/MAP= 342 3.00×10-5) in all population; PCYOX1 (PPWAS-ROS/MAP= 2.70×10-4) in females; and BCAN 343 (PPWAS-ROS/MAP= 8.14×10-4) in males; et al."). What is the et al supposed to mean here?
To the extent that the authors still wish to have their work published in this or any other journal, I believe the article needs to be rewritten from scratch. The idea, the methodology, the results and the conclusion drawn from them must be so much more clearly elaborated. In its revised state, the article seems even further from potential publication than in its original state.
Author Response
Response to Reviewer 2 Comments
Dear editor and reviewers,
We are grateful that you have given us an opportunity to reconsider our revised manuscript entitled “Brain proteome-wide association study identifies candidate genes that regulate protein abundance associated with post-traumatic stress disorder” (Manuscript ID: genes-1639995). We have addressed the comments from the reviewers very carefully. In the following, we detail our responses to the comments and made the significant changes in the revised manuscript. For ease of reviewing, any significant changes in the revised manuscript are flagged in MS Word's “Track Changes” function.
Reviewer Comments and Suggestions for Authors:
In revising their article, the authors could not at all do justice to the criticisms made by me and the other reviewers. I recommend that the authors fundamentally rewrite the article and make a deep linguistic revision of the text.
In many instances it remains completely unclear what is meant by the authors (e.g. p2 l90 "What's more, as a gene-based approach, PWAS aggregates signals from all variants affecting the same protein-coding gene which helps to detect genes with dominant and recessive effects") This sentence does not make any sense to me even after reading it several times.
Parts of the text even seem to be missing (e.g. p5 l341 "In PWAS analysis combined with ROS/MAP proteome reference weights, a total of 341 8 candidate genes associated with PTSD were identified, such as ADK (PPWAS-ROS/MAP= 342 3.00×10-5) in all population; PCYOX1 (PPWAS-ROS/MAP= 2.70×10-4) in females; and BCAN 343 (PPWAS-ROS/MAP= 8.14×10-4) in males; et al."). What is the et al supposed to mean here?
To the extent that the authors still wish to have their work published in this or any other journal, I believe the article needs to be rewritten from scratch. The idea, the methodology, the results and the conclusion drawn from them must be so much more clearly elaborated. In its revised state, the article seems even further from potential publication than in its original state.
Response: Thank you for your helpful comment.
We are sorry for the unclear description. In the process of revising the article, we attached great importance to the comments of reviewers, humbly accepted the criticism from each reviewer, and carefully revised the article according to the comments of each reviewer.
Based on your comments, we have revised the unclear description in our article, and deleted some meaningless description and ambiguous sentence in the introduction section without affecting the understanding of the article, so as to make readers better understand our research.
In the description of the results, we are sorry for the incorrect used of the word "et al" resulted in the absence of the results presentation. In order to show one or two representative and significant candidate genes in different populations and avoid redundant descriptions in the description of the PWAS and TWAS analysis results we used omitted meanings words as the end. Unfortunately, the inappropriate omission may cause misunderstanding. Based on your comments, we have deleted the misused word "et al" in the results section.
We appreciate your valuable suggestions in the revision process of this article. We reviewed and revised our article from scratch, and quoted the most closely and authoritative articles as references to support our research opinions. Moreover, for improving the language, we sent this manuscript to our collaborator, who is a native English speaker. We carefully checked and corrected all the mistakes that we can find. Please see the revised manuscript.
We sincerely request that you reconsider our manuscript. Once again, thank you for all your time and effort in helping us improve and clarify our manuscript.